# The Evolution of Interfacial Microstructure and Fracture Behavior of Short Carbon Fiber Reinforced 2024 Al Composites at High Temperature

**Chi Zhang** [1,2]**, Jinhao Wu** [1,2]**, Qingnan Meng** [1,2,*]**, Youhong Sun** [2,3,*] **and Mao Wen** [4]

1 College of Construction Engineering, Jilin University, Changchun 130026, China
2 Key Lab of Drilling and Exploitation Technology in Complex Conditions, Ministry of Natural Resources, Changchun 130061, China
3 School of Engineering and Technology, China University of Geosciences, Beijing 100083, China
4 Department of Materials Science, Jilin University, Changchun 130026, China
* Correspondence: qingnanmeng@jlu.edu.cn (Q.M.); syh@jlu.edu.cn (Y.S.)

**Abstract:** The short carbon fiber reinforced 2024 Al composites were fabricated through powder metallurgy. The effect of short carbon fiber content on the interfacial microstructure and fracture behavior of the composites at different temperatures were investigated. The results showed that the dislocation accumulation was formed in the aluminum matrix due to the thermal expansion mismatch between carbon fiber and aluminum matrix. With the testing temperature increasing, the size of interfacial product $Al_4C_3$ and precipitates $Al_2Cu$ became larger, and the segregation of $Al_2Cu$ was found coarsening around $Al_4C_3$. The addition of short carbon fiber improved the hardness and modulus of the aluminum matrix in the vicinity of the interface between carbon fiber and aluminum matrix. Compared to the matrix 2024 Al, the yield strength and ultimate tensile strength of the composites first increased and then decreased with increasing short carbon fiber content at room temperature 423 Kand 523 K. The fracture surface of the composites at room temperature was characterized by shear failure of fiber, while the interface debonding and fiber pulled-out became predominant fracture morphologies for the fracture surface at increased temperatures.

**Keywords:** carbon fiber; aluminum matrix composites; interfacial microstructure; mechanical properties; fracture behavior

---

## 1. Introduction

Due to the attractive characteristics of low density combined with high density, aluminum alloys have been used as structural materials in many industries, such as automobiles, aerospace, as well as drill pipe in extra-deep oil drilling [1–4]. However, the deterioration in the mechanical properties of aluminum alloys at high temperature has largely restricted their usability in critical components [5,6]. To overcome the limitations of the high-temperature mechanical properties of aluminum alloys, carbon materials (carbon fiber, carbon nanotube (CNT), graphene, etc.) reinforced aluminum matrix composites (AMCs) have been fabricated owing to their excellent properties such as high strength, high modulus, heat resistance and wear resistance [7–11].

The main problems limiting the application of carbon materials reinforced AMCs are concluded as the poor wettability and in situ chemical reaction between carbon materials and aluminum matrix. Due to non-wetting conditions between monolithic phases, it is difficult to fabricate carbon materials reinforced AMCs with high mechanical properties using traditional casting technologies [12]. To overcome this issue, researchers have proposed various modified processing methods, such as squeeze casting [13,14], liquid infiltration [15–17] and powder metallurgy [18–21]. On the other hand,

the interfacial reaction plays a critical role in the mechanical properties of composites. Due to the low Gibbs free energy of the chemical reaction between aluminum and carbon, aluminum carbide ($Al_4C_3$) is formed at a temperature above 773 K [12,22]. The formation of $Al_4C_3$ has been widely observed in CNTs/Al [23,24], graphene/Al [25,26] and carbon fiber/Al composites [27,28]. However, the effects of $Al_4C_3$ on interfacial bonding and mechanical properties are still under debate. Some researchers claimed that the formation of brittle $Al_4C_3$ is detrimental to the mechanical properties of carbon materials/Al composites [29–31]. Zhang et al. [31] found that the tensile strength of T700 carbon fiber/Al composite decreased by 49% compared to that of M40 carbon fiber/Al composite due to the formation of coarsened $Al_4C_3$. In contrast, other researchers found that the $Al_4C_3$ formation is desirable for improving the interfacial bonding and solving the un-wetting condition [22,32–34]. Song et al. [22] found that nanosized $Al_4C_3$ was beneficial for increasing the interface shear stress and improving the strength of composites, while micro-scale $Al_4C_3$, which was enlarged at high temperature, deteriorated the mechanical properties.

For carbon fiber reinforced metal matrix composites, the addition of carbon fiber could largely change the fracture behavior of composites. Due to the low plasticity of carbon fiber, the elongation for carbon fiber metal matrix composites displayed a significant decrease. On the fracture surface of composites, carbon fiber was usually observed to be damaged in three modes, i.e., fiber pulled-out, debonding and fiber fracture [35], which was determined by the interface bonding between carbon fiber and matrix. The weak interface bonding strength led to the propagation of cracks along the interface, and finally, carbon fiber was pulled out or debonded from the matrix. On the other hand, when the tip stress of crack was lower than the interface bonding strength, carbon fiber tended to be sheared fractured. So far, many researchers have studied the fracture behavior of carbon fiber reinforced metal matrix composites at room temperature. For example, Daoud [28] found that cracks in continuous carbon fiber/2014 Al were mainly generated in the matrix and the strong interface bonding caused shearing of fibers. It is known that temperature played a critical role in affecting the interfacial microstructure of composites. Thus, the fracture behavior of composites might be different at high temperatures.

In this work, the short carbon fiber (SCF) reinforced 2024 Al composites were fabricated by mixing 2024 Al powders with SCFs and vacuum hot pressing. The influence of SCFs addition on the mechanical properties of SCFs/2024 Al composites was investigated. The evolution of interfacial microstructure and fracture behavior of SCFs/2024 Al composites at high temperature were discussed in detail.

## 2. Materials and Methods

T700 polyacrylonitrile (PAN)-based carbon fiber was provided by Toray Co. Ltd., Tokyo. The density of the PAN-based carbon fiber was 1.80 g/cm$^{-3}$, and the average diameter was 7 μm. The PAN-based carbon fiber was immersed in acetone for 4 h to remove sizing agent, and then desized into SCF by ball milling. The final aspect ratio of SCF was less than 42.8. The as received 2024 Al powder (Chaowei Nanotechnology Co. Ltd., Shanghai, China) was ~20 μm in D50 particle size and 2.78 g/cm$^{-3}$ in density. The composition of the matrix was: 0.125 wt % Fe, 0.087 wt % Si, 3.952 wt % Cu, 0.450 wt % Mn, 1.283 wt % Mg, 0.137 wt % Zn and the balance aluminum.

SCFs were uniformly dispersed in 2024 Al powder through mechanical mixing using ethyl alcohol as mixing agent. The blended powders were then placed in a graphite mold with Φ60 mm diameter and vacuum dried at 343 K for 2 h to remove ethyl alcohol. SCFs reinforced 2024 Al matrix composites were fabricated by heating in a vacuum furnace at 853 K for 40 min, during which a 30 MPa pressure was applied on the powders. The SCFs/2024 Al were then T6-treated (solution treated at 768 K for 60 min, water quenched and then aged at 463 K for 12 h). Besides, the comparative sample 2024 Al was prepared by powder metallurgy (PM) using the same method.

The microstructures were characterized by a scanning electron microscopy (SEM.; S-4800, Tokyo, Japan) and a transmission electron microscopy (TEM, Titan G2 60–300, Hillsboro, USA) equipped

with scanning transmission electron microscopy (STEM) and energy dispersive spectrometer (EDS). The phase constituents of samples were identified by X-ray diffraction (XRD, D/Max 2500 PC, Tokyo, Japan) using Cu Kα radiation in step mode from 20° to 80°. The density of SCFs/2024 Al was measured through Archimedes' principle [36]. The specimens for the tensile test were prepared in accordance with ASTM Standard E-8/E8M-09 with a 10 mm gauge length. Tensile tests were performed at a strain rate of $8.3 \times 10^{-4}$ s$^{-1}$ on INSTRON-1121 material testing machine at room temperature 423 and 523 K. The Vickers hardness was measured via a microhardness tester (1600–5122 VD Microment 5104, Chicago, USA) under an applied load of 100 g for 15 s. At least seven measurements were performed to ensuring the accuracy of results. Further, nanoindentation measurement was done using an MTS XP nanoindentation with a Berkovich type pyramidal diamond indenter. The indent with 2 μm spacing between neighboring indents and 200 nm depths were used. The load-displacement curve of the indentation was obtained by recording the applied forces on the indenter and the corresponding displacements. The contact hardness H was determined by

$$H = \frac{P_{max}}{A_c} \tag{1}$$

where $P_{max}$ and $A_C$ was the peak load and the real contact area, respectively.

The elastic modulus $E$ was calculated using the following equation:

$$\frac{1}{E_r} = \frac{1 - v^2}{E} + \frac{1 - v_i^2}{E_i} \tag{2}$$

where $E_i$ was the modulus of the indenter, $v$ and $v_i$ were the Poisson's ratio for the specimen and the indenter, respectively. The values $E_i$ = 1141 GPa, $v_i$ = 0.07 [37] for the indenter, $v$ = 0.33 for the aluminum matrix and $v$ = 0.30 for the carbon fiber [38] were used in all computations. The effective elastic modulus, $E_r$, which accounts for elastic displacements in both the specimen and the indenter, was evaluated from

$$E_r = \frac{1}{\beta} \frac{\sqrt{\pi}}{2} \frac{S}{\sqrt{A_c}} \tag{3}$$

where $\beta$ was a correction factor, which depends on the indenter geometry. For Berkovich indenter, $\beta$ was a constant with a value of 1.034 [39].

## 3. Results and Discussion

In this study, the SCFs were added in 2024 Al matrix with volume fraction ranging from 2 to 8%. The SEM morphology for the SCFs/2024 Al are presented in Figure 1. As shown in Figure 1, SCFs are dispersed uniformly in the 2024 Al matrix with no obvious clustering being observed. It suggests that the clustering of SCFs could be avoided by reducing the aspect ratio. It is also clear that no pore or cracking is found between SCF and matrix.

The density of 2024 Al against different volume fractions of SCFs/2024 Al is displayed in Figure 2a. As shown in Figure 2a, the addition of SCFs results in a lower density of SCFs/2024 Al than that of 2024 Al matrix (2.763 g/cm$^{-3}$). In the meantime, with the volume fraction of SCFs increasing from 2 to 8 vol. %, the density decreases from 2.746 to 2.692 g/cm$^{-3}$. The theoretical density of the composite is determined by

$$\rho_{theoretical} = \rho_C V_C + \rho_M V_M \tag{4}$$

where $\rho_C$ (1.80 g/cm$^{-3}$) and $\rho_M$ (2.78 g/cm$^{-3}$) are the theoretical density of carbon fiber and 2024 Al matrix, respectively. $V_C$ and $V_M$ are the volume fraction of carbon fiber and 2024 Al matrix, respectively. Therefore, the theoretical density for the composites with 2, 4, 6 and 8 vol. % is 2.760, 2.741, 2.721 and 2.702 g/cm$^{-3}$, respectively. The relative density $\rho_{relative}$ is determined by

$$\rho_{relative} = \rho_{measured} / \rho_{theoretical} \tag{5}$$

where $\rho_{measured}$ is the measured density. The relative density of SCFs/2024 Al and 2024 Al matrix is shown in Figure 2b. All composites with different fractions of SCFs achieve above 99% in relative density, indicating that low porosity is achieved through powder metallurgy.

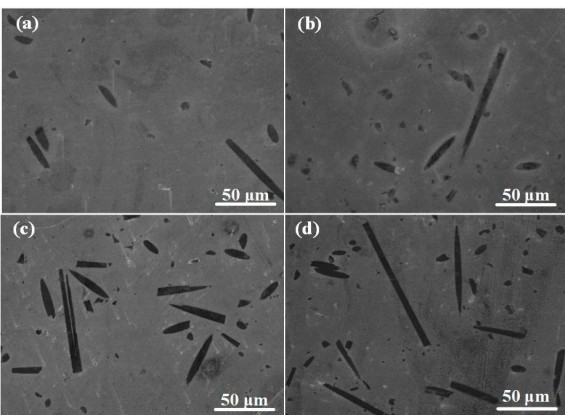

**Figure 1.** SEM images for the 2024 Al matrix composites with (**a**) 2, (**b**) 4, (**c**) 6 and (**d**) 8 vol. % short carbon fibers (SCFs).

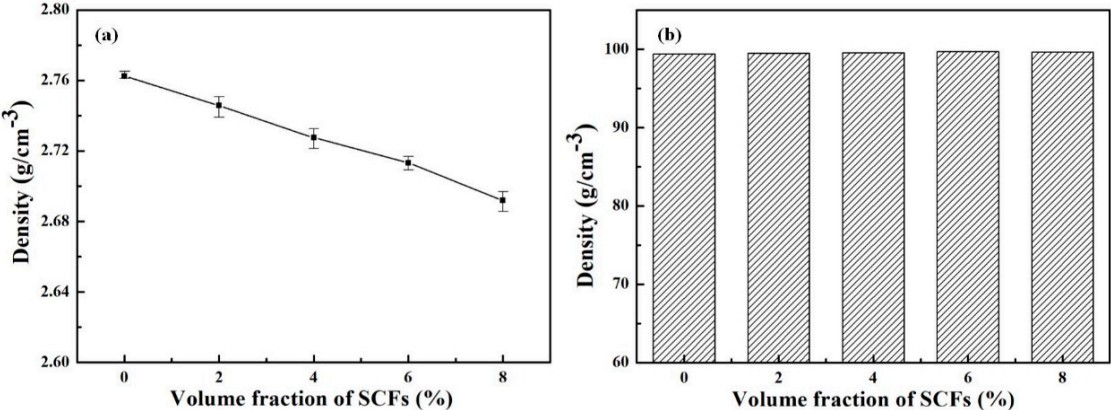

**Figure 2.** The (**a**) measured density and (**b**) relative density for the composites with different volume fraction of SCFs.

Figure 3a presents the XRD patterns for 8 vol. % SCFs/2024 Al composite and 2024 Al. According to the XRD results, major aluminum peaks are detected at 38.4° (1 1 1), 44.8° (2 0 0), 65.1° (2 2 0) and 78.3° (3 1 1) (PDF#65–2869) for 2024 Al. In contrast, the four aluminum peaks detected in the XRD pattern for 8 vol. % SCFs/2024 Al composite all shift ~0.2° to a higher value, indicating aluminum grains in the SCFs/2024 Al composite are subjected to tensile strain. Owing to the sensitivity of XRD, no peaks representing typical precipitates (such as $Al_2Cu$) or interfacial product (such as $Al_4C_3$) are observed in the XRD patterns for both 2024 Al and SCFs/2024 Al composites. It is worth noting that no carbon peak representing carbon fiber is detected in the XRD pattern for SCFs/2024 Al composite with the highest SCFs content (8 vol. %). Seong [27] reported that PAN-based carbon fibers are largely amorphous, as evidenced by the relative lack of preferential orientation in the graphite basal planes. Figure 3b shows the average crystallite size obtained based on XRD patterns and Scherrer's equation [40]. 2024 Al has the largest average crystallite size with 81 nm. The addition of SCFs results in a decrease in crystallite size of aluminum matrix, and the crystallite size decreases with the increasing SCFs content.

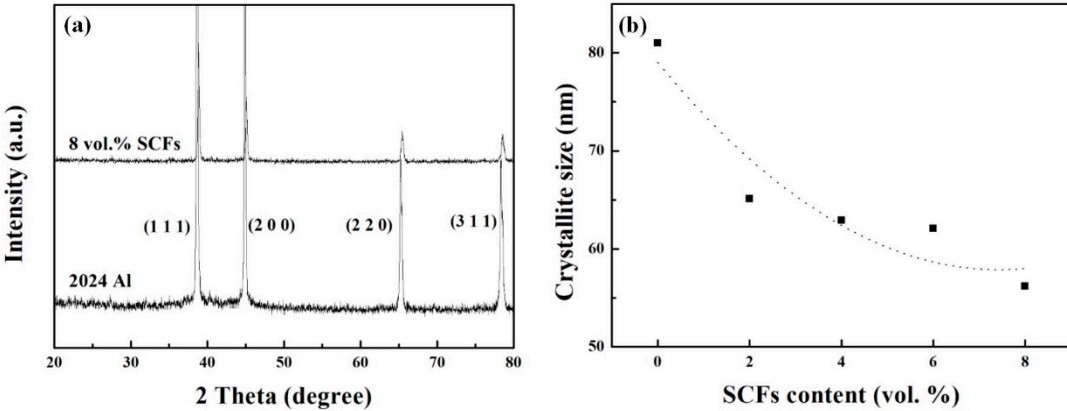

**Figure 3.** (**a**) XRD patterns for 2024 Al and 8 vol. % SCFs/2024 Al composites, (**b**) the relationship between crystallite size of the aluminum matrix and SCFs content.

Figure 4 illustrates the morphology of the interface between carbon fiber and matrix in 4 vol. % SCFs/2024 Al composite. As shown in Figure 4a, different from the XRD results, the interface is rich in the rod-like interfacial products (highlighted by white arrows), which have an average size of 28 nm in width. The interfacial products nucleate discontinuously at the fiber surface and grow into the matrix in various directions. Meanwhile, Figure 4b shows that large amounts of dislocations are accumulated in the matrix, especially in the vicinity of carbon fiber. These dislocations are mainly formed due to the large thermal expansion mismatch between carbon fiber and matrix during the cooling period. Figure 4c displays the magnified morphology of the interfacial product embedding in the interface between carbon fiber and matrix. Figure 4d shows the high resolution transmission electron microscopy (HRTEM) image of the selected area in Figure 4c. The HRTEM image shows that lattice fringes with a spacing of 0.83 nm lie parallel to the crystal axis. It is reported that the $Al_4C_3$ parallel to the c-axis is three fringe spacings, or 2.49 nm [41]. Furthermore, the inserted fast Fourier transformation (FFT) pattern confirms the compounds as $Al_4C_3$, which is in hexagonal structure with lattice parameters of a = 0.33 nm and c = 2.49 nm [41].

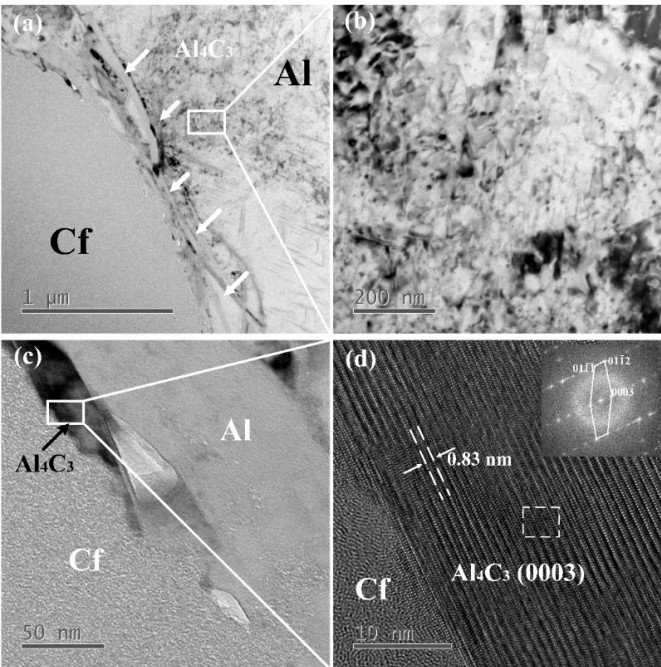

**Figure 4.** (**a**) Bright-field transmission electron microscopy (TEM) morphologies of the fiber/matrix interface in the SCFs/2024 Al composites, (**b**) enlarged view of the dislocation accumulation in the matrix, (**c**) enlarged view of the $Al_4C_3$ embedding at the interface and (**d**) corresponding HRTEM image of the selected area in Figure 4c.

Figure 5 presents the TEM micrograph of θ′ (Al₂Cu) precipitates in the aluminum matrix. The plate-shaped θ′ precipitates are observed along <101> direction of the α-Al matrix, and its size is about 90 nm in length. Figure 6a shows the interface morphology between $Al_4C_3$ and α-Al matrix. It can be seen that white precipitates are formed at the interface between $Al_4C_3$ and α-Al matrix. From the EDS mapping results (Figure 6b–d), these precipitates are rich in aluminum and copper, indicating the segregation of θ′ (Al₂Cu) surrounding $Al_4C_3$.

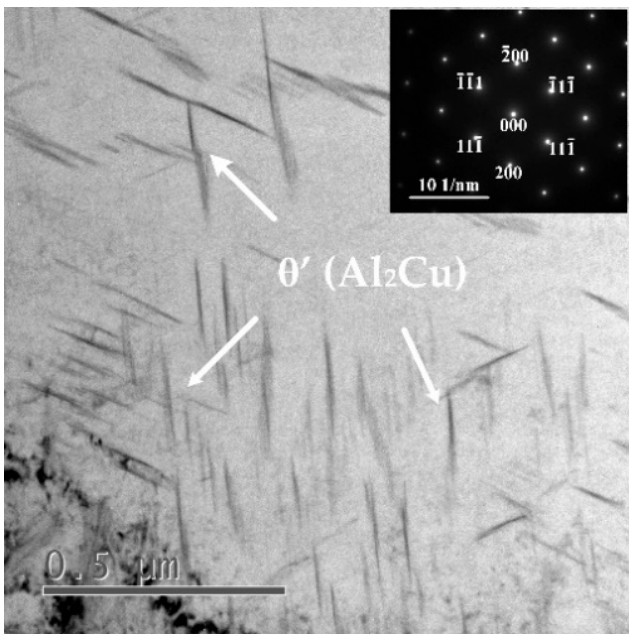

**Figure 5.** The TEM morphology of θ′ (Al₂Cu) precipitates along <101> direction of α-Al matrix.

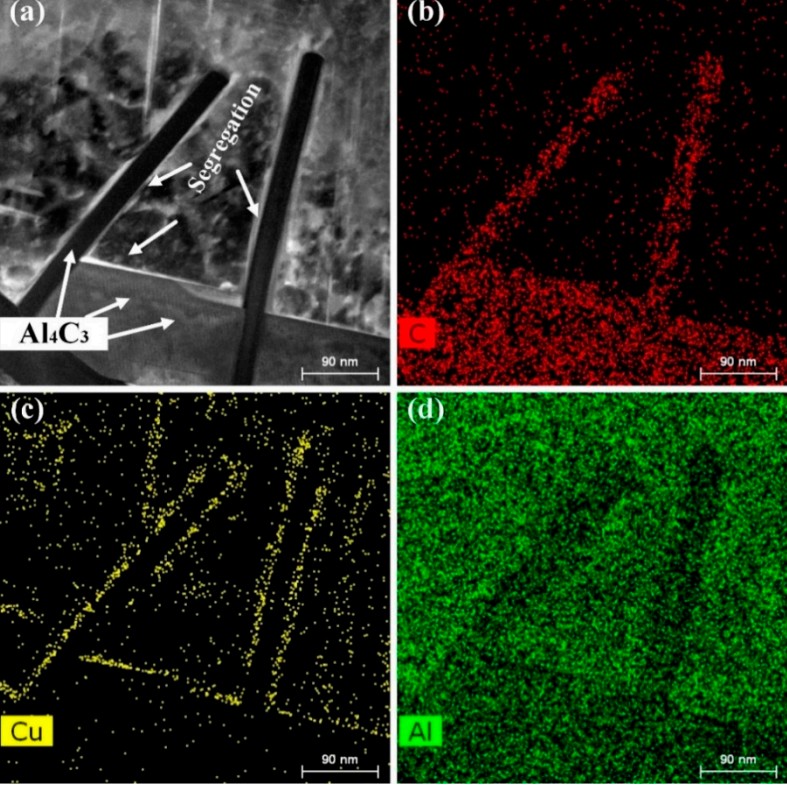

**Figure 6.** (**a**) High angle annular dark field (HADDF) STEM image (Z-contrast) for the interfacial microstructure of the SCFs/2024 Al composite, (**b–d**) EDS mapping analysis for Figure 6a.

The Vickers hardness results of 2024 Al and SCFs/2024 Al are presented in Figure 7. 2024 Al has the lowest hardness (127.8 HV). The Vickers hardness increases with the increasing SCFs content. The samples for 8 vol. % SCFs/2024 Al composite possess the highest Vickers hardness (160.6 HV), which is 25.6% higher than that of 2024 Al. It is worth noting that a wider error range is obtained for the composites with higher volume fractions of SCFs. The inset in Figure 7 displays the indentation for the measured point with the highest hardness (170.1 HV). It can be seen that the indentation covers several SCFs. It is obvious that the indenter is more likely to press on the zone around SCFs with the increasing volume fraction of SCFs, and thus, results in the fluctuation in Vickers hardness for the composites with high volume fractions of SCFs.

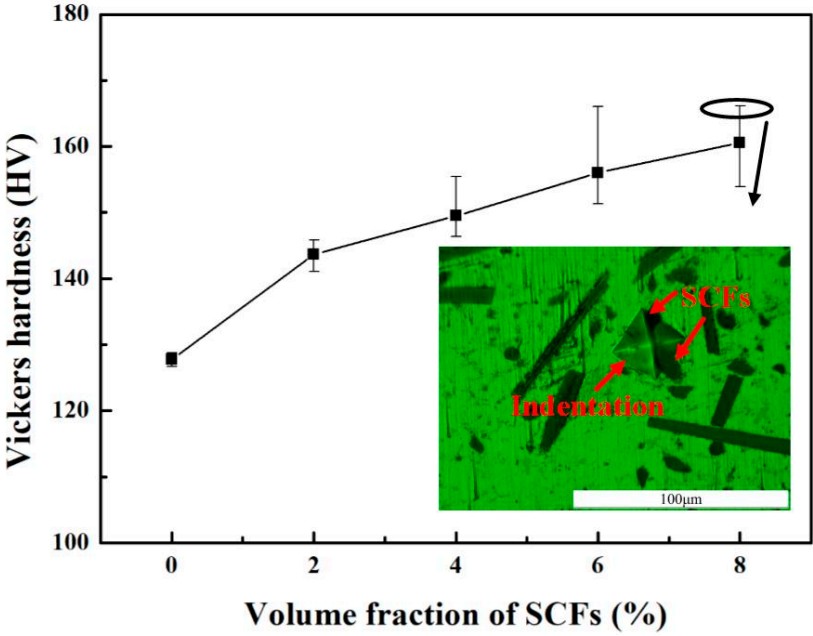

**Figure 7.** The variation tendency of Vickers hardness with the volume fraction of SCFs, and the inset shows the indentation with high hardness.

In order to further study the influence of SCFs on the hardness of the 2024 Al matrix, naonindentation was performed on a single SCF and the matrix around it. The indentation load-displacement curves for the SCF and matrix are shown in Figure 8a, in which both the indentation tests have a fixed maximum displacement of 200 nm. The peak load of SCF (2.31 mN) is much higher than that of the 2024 Al matrix (1.65 mN), indicating that the SCF possesses a higher hardness than the nearby matrix. Figure 8b displays the hardness (H) and modulus (E) variation as indentation moving from a single SCF to the aluminum matrix. The SCF has the highest hardness at 4.58 GPa on average and lowest modulus at 28.32 GPa on average. For the aluminum matrix near the single SCF, the indentation in the vicinity of carbon fiber is provided with highest modulus (93.53 GPa) and hardness (2.51 GPa). As the indentation moves far away from the carbon fiber, the modulus and hardness for the matrix both decrease. It indicates that in SCF is able to enhance the hardness and modulus of the nearby matrix, and the enhancement weakens with the increasing distance from the 2024 Al matrix to the fiber.

Figure 9a–c shows the engineering stress versus strain curves for 2024 Al and 2–8 vol. % SCFs/2024 Al composites at room temperature, 423 K and 523 K respectively, while Figure 9d–f and Table 1 summarize the typical tensile properties for all samples, including yield strength (YS), ultimate tensile strength (UTS) and fracture strain ($\varepsilon_f$). The mechanical strengths for 2024 Al and SCFs/2024 Al composites display a sound dependence on temperature and SCFs content. The increasing temperature gives rise to the deterioration in mechanical strengths for 2024 Al and SCFs/2024 Al composites. The YS and UTS of the composites first increase and then decrease with an increasing volume fraction of SCFs.

At all temperatures, the samples for 4 vol. % SCFs/2024 Al composites possess the highest YS and UTS. In comparison with 2024 Al, the YSs for 4 vol. % SCFs/2024 Al composites are 19.6%, 45.6%, and 36.7% higher than those of 2024 Al at room temperature, 423 K and 523 K, while the UTSs increase 4.6%, 13.7%, and 16.2%, respectively. It is worth noting that the addition of SCFs results in the significant decrease in elongation for SCFs/2024 Al composites. From the engineering stress-strain curves for 2024 Al and SCFs/2024 Al composites (Figure 9a), unlike the apparent plastic deformation for 2024 Al, SCFs/2024 Al composites are found fractured immediately after yielding at room temperature. Meanwhile, with the temperature increasing, the engineering stress-strain curves for SCFs/2024 Al (Figure 9b,c) exhibit a relatively obvious plastic deformation after yielding. It indicates that temperature plays a critical role in affecting the fracture behavior of SCFs/2024 Al composites.

The fracture morphologies of 4 vol. % SCFs/2024 Al composites at different temperatures are displayed in Figure 10. For the fracture surface of 4 vol. % SCFs/2024 Al composite at room temperature, the existence of dimples is observed on the 2024 Al matrix, indicating the ductile failure for the 2024 Al matrix. The primary reason causing the dramatic decrease in fracture strain is the addition of SCFs. It is worth noting that the debonding between SCF and aluminum matrix is hard to observe, and most of the SCFs are subjected to shear failure (as shown in Figure 10a,b). At 423 K, the amount of sheared SCFs decreases. Meanwhile, the debonding between SCFs and matrix is existent, and some SCFs are observed to be pulled out from the matrix (Figure 10c,d). When the temperature increases to 523 K, the predominant fracture morphologies for the composite debond and pull-out SCFs (Figure 10e,f).

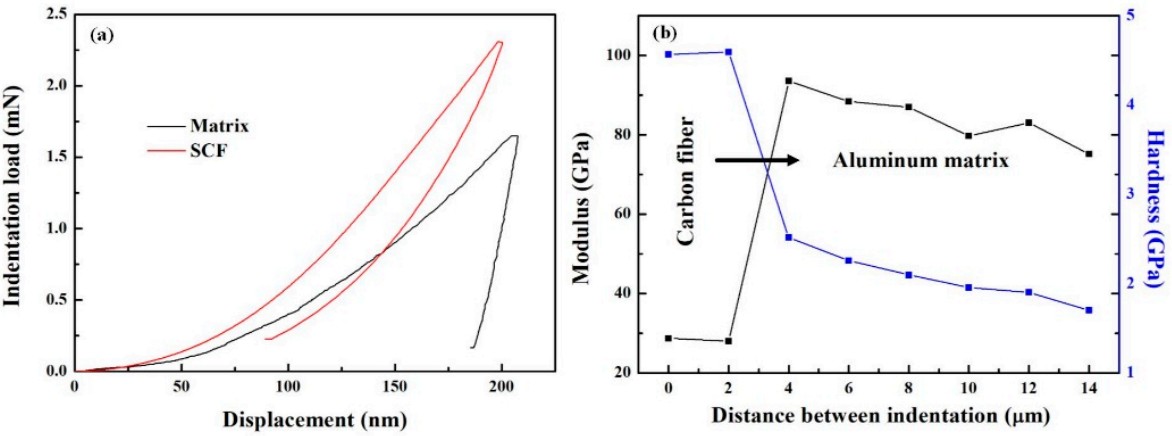

**Figure 8.** (**a**) Typical load-displacement curves of nanoindentation tests of the SCF and the matrix, (**b**) variation of indentation hardness and modulus of the carbon fiber and adjacent aluminum matrix.

**Table 1.** The tensile tests data of 2024 Al and the composites at different temperatures.

| Samples | ROOM Temperature | | | 423 K | | | 523 K | | |
|---|---|---|---|---|---|---|---|---|---|
| | YS/MPa | UTS/MPa | $\varepsilon_f$/% | YS/MPa | UTS/MPa | $\varepsilon_f$/% | YS/MPa | UTS/MPa | $\varepsilon_f$/% |
| 2024 Al | 32 ± 11 | 414 ± 1 | 11.8 ± 0.6 | 219 ± 4 | 350 ± 5 | 11.6 ± 0.2 | 188 ± 10 | 246 ± 2 | 5.8 ± 0.2 |
| 2 vol. % | 368 ± 5 | 423 ± 6 | 5.9 ± 0.7 | 291 ± 3 | 384 ± 10 | 5.4 ± 0.1 | 252 ± 1 | 286 ± 2 | 4 ± 0.2 |
| 4 vol. % | 384 ± 4 | 433 ± 2 | 5.2 ± 0.4 | 319 ± 5 | 398 ± 6 | 5 ± 0.2 | 257 ± 5 | 291 ± 4 | 3.9 ± 0.6 |
| 6 vol. % | 357 ± 11 | 380 ± 9 | 4.3 ± 0.3 | 251 ± 6 | 326 ± 7 | 4.7 ± 0.2 | 244 ± 1 | 276 ± 3 | 3.3 ± 0.2 |
| 8 vol. % | 343 ± 8 | 360 ± 8 | 2.8 ± 0.6 | 248 ± 3 | 283 ± 6 | 3.6 ± 0.1 | 242 ± 2 | 272 ± 4 | 3 ± 0.2 |

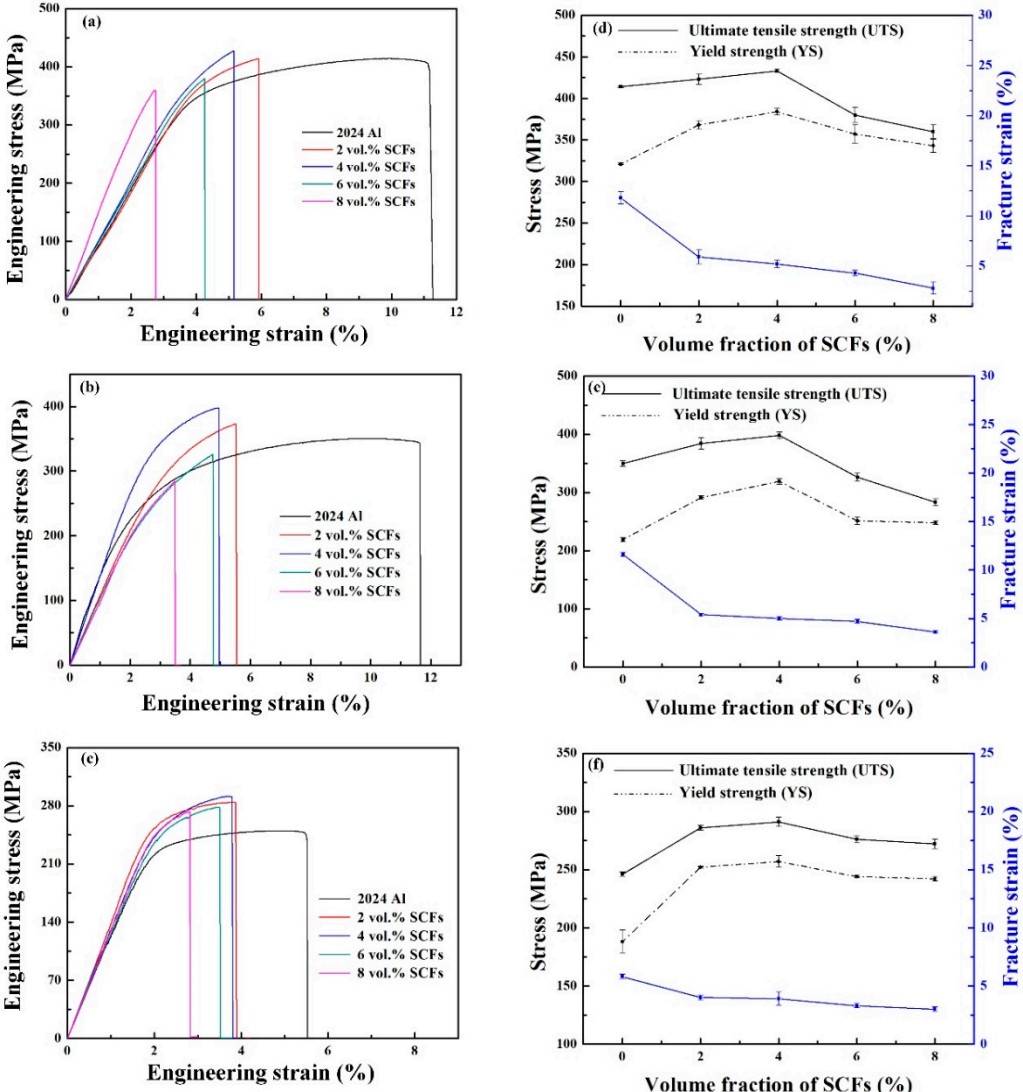

**Figure 9.** Engineering stress-strain curves and evolution curves for the tensile properties of 2024 Al and SCFs/2024 Al composites at (**a**) and (**d**) room temperature, (**b**) and (**e**) 423 K and (**c**) and (**f**) 523 K.

Figure 11 presents the TEM micrographs for the 4 vol. % SCFs/2024 Al composites after the tensile test at 523 K. As shown in Figure 11a, after the tensile test at 523 K, the composite has a larger density of $Al_4C_3$ at the interface. Meanwhile, the interfacial product has an average size of 70 nm in width, which is larger than the size of $Al_4C_3$ at the interface in the composite at room temperature (as shown in Figure 4a). The plate-shaped θ' ($Al_2Cu$) precipitates along <101> direction of α-Al matrix also have a larger average size of ~150 nm (Figure 11b), indicating the coarsening of precipitates at high temperature (523 K). Figure 11c shows that the dislocation accumulation takes place in the Al matrix near the interface between carbon fiber and 2024 Al. The STEM and EDS mapping results reveal that large amounts of $Al_2Cu$ segregation form in the area around $Al_4C_3$, and the segregation coarsens a lot compared to that in SCFs/2024 Al composites at room temperature (as displayed in Figure 6).

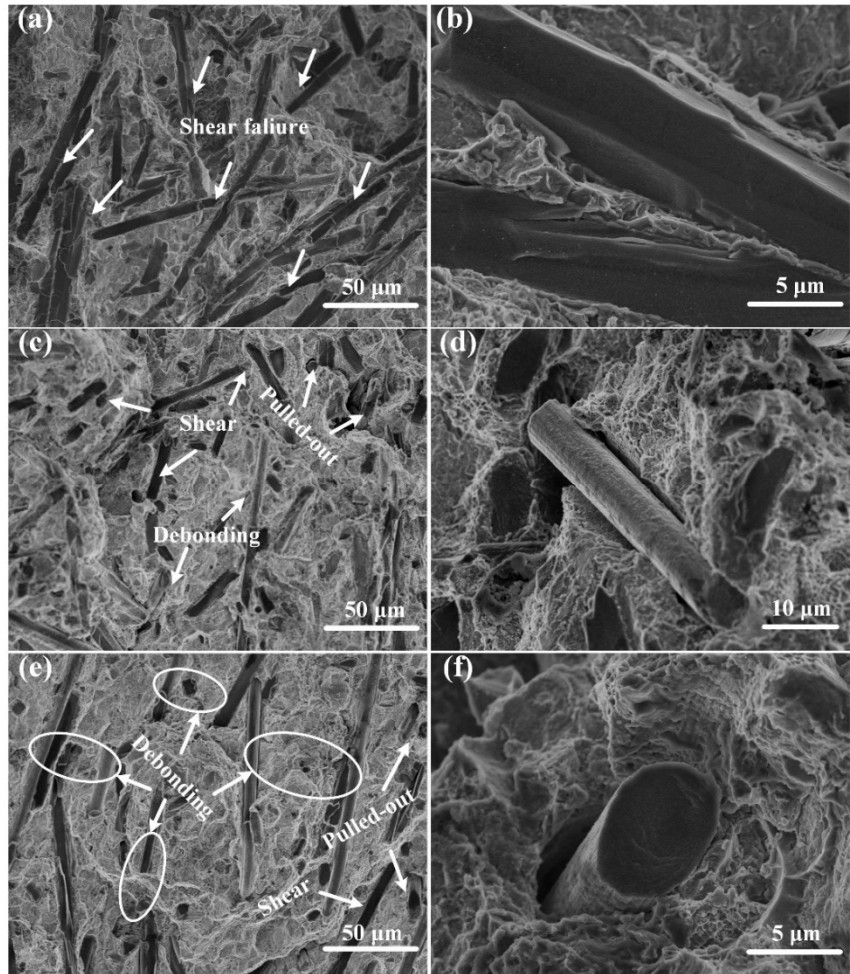

**Figure 10.** The SEM images for the fracture surface of 4 vol. % SCFs/2024 Al composites at (**a**,**b**) room temperature, (**c**,**d**) 423 K and (**e**,**f**) 523 K.

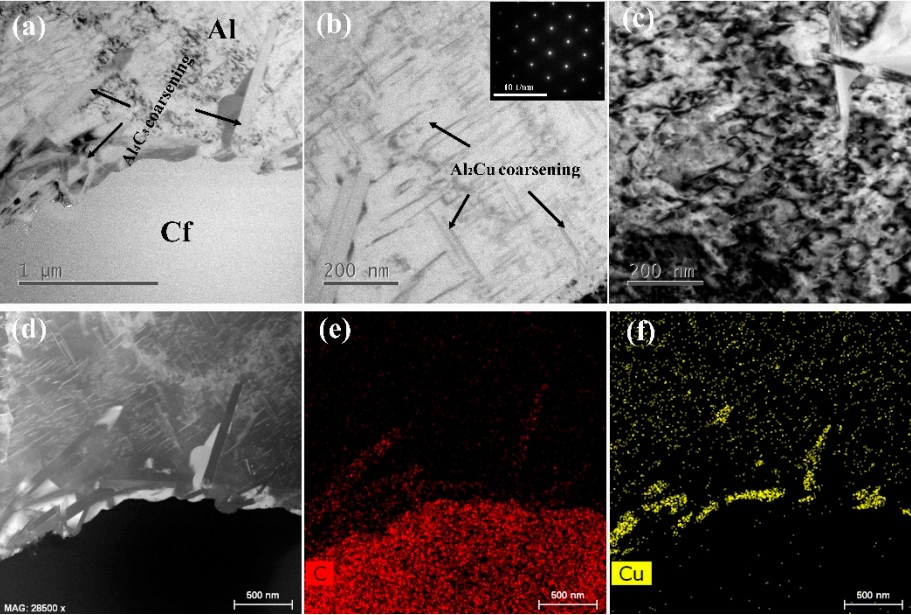

**Figure 11.** (**a**–**c**) TEM micrographs, (**d**) HADDF STEM micrograph and (**e**,**f**) corresponding EDS mapping results for the 4 vol. % SCFs/2024 Al composite after tensile test at 523 K.

From the results of Vickers hardness tests (Figure 7), the addition of SCFs gives rise to the significant improvement in Vickers hardness of SCFs/2024 Al composites, and the Vickers hardness increases with the increasing volume fraction of SCFs. Corresponding to the results of nanoindentation (Fig 8a), carbon fiber, as a hard phase, processes much higher hardness than 2024 Al. From the view of the rule of mixture, the increasing volume fraction of SCFs is able to cause higher hardness of SCFs/2024 Al composites. On the other hand, the addition of SCFs results in the decrease in the crystallite size of the aluminum matrix (Figure 3b), which indicates that SCFs could efficiently impede the coarsening of aluminum grains during the thermal process. According to Hall–Petch relationship, grain refinement creates more grain boundaries, and the high free energy at grain boundaries is able to resist the dislocation motion [42]. Meanwhile, dislocation accumulation is observed in the aluminum matrix near the interface (Figure 4b), resulting in the strengthening of aluminum matrix and leading to a higher hardness, which is in agreement with the results of nanoindentation (Figure 8b).

The results of tensile tests show that SCFs give rise to the improvement in yield strength of the SCFs/2024 Al composites compared to that of 2024 Al. At all three temperatures, the yield strength of the composites first increases and then decreases with the SCFs content increasing. Due to the thermal expansion mismatch between carbon fiber and aluminum matrix, residual stress is generated at the interface. Meanwhile, residual stress is able to cause the dislocation accumulation in the matrix near the interface (as shown in Figure 4b). It is important to note that the dislocation accumulation observed at the interface plays a key role in improving the yield strength of SCFs/2024 Al. In addition, the residual stress can be calculated in accordance with the equation below [43]:

$$\sigma_{CF} = (\alpha_{CF} - \alpha_m)\Delta T E_{CF} \tag{6}$$

where $\sigma_{CF}$ is the stress carbon fiber suffered in radial direction, $\alpha_{CF}$ and $\alpha_m$ is the thermal expansion coefficient of carbon fiber in radial direction ($7.2 \times 10^{-6}$ K$^{-1}$ [35]) and aluminum matrix ($23.6 \times 10^{-6}$ K$^{-1}$ [36]), $\Delta T$ is the difference between sintering temperature and room temperature, and $E_{CF}$ is the modulus of carbon fiber. Due to the residual stress, the carbon fiber in composites are subjected to compressive stress, as $\alpha_{CF}$ is much smaller than $\alpha_m$. Reversely, the aluminum matrix is subjected to tensile stress, which is in agreement with the shifted aluminum peaks observed in the XRD patterns for SCFs/2024 Al composites (Figure 3a). The inner tensile stress is adhesive to increasing the yield strength for SCFs/2024 Al. With the SCFs content increasing, the inner tensile stress becomes the predominant disadvantage and causes the deduction in strength. At high temperature (423 K and 523 K in present work), 2024 Al and SCFs/2024 Al composites all sustain an obvious decrease in tensile strength, because of the coarsening of Al$_2$Cu phase (as shown in Figure 10b). However, the residual dislocations in the matrix (as shown in Figure 11c) ensure the relatively higher strength of SCFs/2024 Al composites compared to that of 2024 Al.

The addition of SCFs results in the deduction in fracture strain for SCFs/2024 Al composites. As shown in Figure 10a,b, the fracture surface of the SCFs/2024 Al composite at room temperature is mainly characterized by the shear failure of carbon fiber. This indicates the strong interfacial bonding between carbon fiber and aluminum matrix. The formation of Al$_4$C$_3$ is observed at the interface between carbon fiber and aluminum matrix by TEM characterization (Figure 4a). Al$_4$C$_3$ is reported to have negative and positive effects on the interfacial bonding between aluminum matrix and carbon materials. On one hand, the formation of nano-sized Al$_4$C$_3$ could increase the wettability between aluminum and carbon materials and enhance the interfacial bonding [22,33]. On the other hand, a larger size of Al$_4$C$_3$ is adhesive to the interfacial bonding due to its intrinsic brittleness [31]. It is worth noting that due to the residual stress generated at the interface during the cooling process, carbon fiber is in compress. The interfacial residual stress in radial direction remarkably strengthens the interfacial bonding and increases the friction between carbon fiber and aluminum matrix, which makes the carbon fiber difficult to be pulled out. Therefore, the cracks propagate towards carbon fiber and cause stress concentration at the tip of cracks, and finally result in the shear failure of carbon fiber. In contrast, the SCFs/2024 Al composite after the tensile test at high temperature exhibits a

different fracture morphology. The amount of debonding and pulled-out carbon fibers are obviously increasing, revealing the weakening in interfacial bonding between carbon fiber and aluminum matrix. The deterioration in interfacial bonding is mainly caused by the following three key factors: The decreasing residual stress at the interface due to the increasing temperature, the coarsening of $Al_4C_3$ (Figure 11a) and segregation (Figure 11d–f). It is generally known that the interfacial debonding is caused by crack propagation. When the interfacial bonding strength is lower than the fracture strength of carbon fiber and the strength at the tip of cracks, the cracks are more likely to deflect along the fibers at the interface. The fibers debonding and pulled-out occur due to the growth of cracks at the interface. Therefore, the fracture behavior for SCFs/2024 Al composites shows much difference at high temperature. At room temperature, carbon fibers are mainly fractured in shear failure because of the strong interfacial bonding induced by residual stress. At high temperature, carbon fibers are more likely to be debonded and pulled out from the matrix, owing to the weakening in interfacial bonding.

## 4. Conclusions

The SCFs reinforced 2024 Al composites were fabricated by powder metallurgy. The effect of SCFs on the interfacial microstructure and mechanical properties of SCFs/2024 Al composites at different temperatures were investigated. The results obtained are summarized as follows:

(1) The dislocation accumulation is formed in the aluminum matrix due to the thermal expansion mismatch between carbon fiber and the aluminum matrix. With the testing temperature increasing, the size of interfacial product $Al_4C_3$ and precipitates $Al_2Cu$ becomes larger, and the segregation of $Al_2Cu$ is found coarsening around $Al_4C_3$.

(2) From the result of nanoindentation, carbon fiber improves the hardness and modulus of the adjacent aluminum matrix. The indentation on the aluminum matrix nearest carbon fiber possesses the highest modulus (93.53 GPa) and hardness (2.51 GPa).

(3) The addition of SCFs gives rise to the sharp decrease in the fracture strain of SCFs/2024 Al composites. The tensile strengths of the composites first increase and then decrease with the increasing volume fraction of SCFs. At room temperature, 423 and 523 K, the highest yield strengths are obtained by 4 vol. % SCFs/2024 Al, which are 19.6%, 45.6%, and 36.7% higher than those of 2024 Al, respectively.

(4) The fracture surface of the SCFs/2024 Al composites at room temperature is mainly characterized by the sheared fracture of carbon fiber, due to the strong interfacial bonding between carbon fiber and the aluminum matrix caused by the residual stress generated at the interface during the cooling process. With the temperature increasing, the interfacial bonding is weakened because of the decreasing residual stress and the coarsening of interfacial product $Al_4C_3$ and $Al_2Cu$ segregation. Therefore, the interface debonding and fiber pulled-out become predominant fracture morphologies for the fracture surface obtained at 523 K.

**Author Contributions:** Q.M. and Y.S. designed the experiment. C.Z. carried out sample preparation and TEM analysis. J.W. and carried out the XRD and SEM analysis. M.W. carried out the mechanical properties' measurements, and C.Z. wrote the paper. All of the authors discussed the data and commented on the paper.

**Funding:** This work was financially supported by the Natural Science Foundation of China (No. 41872181 and 41502344).

**Conflicts of Interest:** The authors declare no conflict of interest.

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
