# Peer review of "The Evolution of Interfacial Microstructure and Fracture Behavior of Short Carbon Fiber Reinforced 2024 Al Composites at High Temperature"

_applsci, doi:10.3390/app9173477_

Round 1

Reviewer 1 Report

The mechanical properties of an Al-matrix composite reinforced with C fibers have been investigated.

Despite the investigation has been performed in a very extensive way and many experimetal results are presented, some points are still critical:

the novelty of the study, all the results presentend are well known and have already been described in literature

the mechanical properties at high temperature are not described in detail, and creep has not even been cited

images should be discussed in greater detail, especially those from TEM. Authors should  use more symbols or text direcly within the pictures in order to improve their comprension

Author Response

Aug. 14, 2019

Applsci_567818

Title: The evolution of interfacial microstructure and fracture behavior of short carbon fiber reinforced 2024 Al composites at high temperature

Dear Editor and reviewer,

Thank you for the valuable comments. We have addressed all comments from the reviewer as follows:

the novelty of the study, all the results presentend are well known and have already been described in literature

Reply: Thanks for your reminding. Short carbon fiber reinforced aluminum matrix composites have been reported in many literatures, and many researchers have conducted systematic study on the properties. We summarized the literatures in this field and listed as follows [1-15]. In the past years, the main focus in this field is on studying the wear and friction behavior, thermal properties and room temperature mechanical properties of SCFs/Al. Various preparation methods were involved here, such as powder metallurgy [5, 10], squeeze casting [7, 11-14], infiltration [1-3, 6, 8], stir casting [4, 15] and thixomixing method [9]. Many researchers found that the addition of SCFs could effectively improve wear and friction properties (room [2, 3, 5, 7, 9, 13] and high temperature [1]), thermal properties [6, 8, 10], and room temperature mechanical properties (hardness [4, 12, 14, 15], tensile strengths [4, 7, 14, 15] and compressive strengths[12]).

In consideration of the engineering background, our group is working on developing aluminum alloy matrix composite drill pipe, which could be applied in deep and extra-deep drilling. The conventional aluminum alloy drill pipe (2024 Al) can hardly work under high temperature (>423 K). Thus, we tried to prepared SCFs/2024 Al composites and studied the high temperature mechanical properties. We also noticed that temperature has influence on the interfacial microstructures and affects tensile fracture behavior. Thus, we thought that the main novelty of the work in focusing on the evolution of interfacial microstructures and tensile fracture behavior at high temperature.

[1]   Yao-hui, L.; Jun, D.; Si-rong, Y.; Wei, W., High temperature friction and wear behaviour of Al2O3 and/or carbon short fibre reinforced Al–12Si alloy composites. Wear 2004, 256, (3-4), 275-285.

[2]    Du, J.; Liu, Y.; Yu, S.; Li, W., Effect of heat-treatment on friction and wear properties of Al2O3 and carbon short fibres reinforced AlSi12CuMgNi hybrid composites. Wear 2007, 262, (11-12), 1289-1295.

[3]    Liu, L.; Li, W.; Tang, Y.; Shen, B.; Hu, W., Friction and wear properties of short carbon fiber reinforced aluminum matrix composites. Wear 2009, 266, (7-8), 733-738.

[4]    Shi, J. C.; Li, Y. H.; Yao, G. C.; Yan, P. F.; Liu, H., Effect of Mg on Microstructure and Mechanical Properties of Copper-Coated Short Carbon Fiber Reinforced Al Alloy Matrix Composite. Advanced Materials Research 2012, 457-458, 348-353.

[5]    Deng, X. Y.; Zhang, G. H.; Qiang, C. W.; Cairang, Z. M.; Wang, T., Friction and Wear Properties of Aluminum Matrix Composites Reinforced by Coated Carbon Fibers. Advanced Materials Research 2013, 684, 342-346.

[6]    Liu, T.; He, X.; Zhang, L.; Liu, Q.; Qu, X., Fabrication and thermal conductivity of short graphite fiber/Al composites by vacuum pressure infiltration. Journal of Composite Materials 2013, 48, (18), 2207-2214.

[7]    Łągiewka, M., Mechanical and Tribological Properties of Metal Matrix Composites Reinforced with Short Carbon Fibre. Archives of Metallurgy and Materials 2014, 59, (2), 707-711.

[8]    Liu, T.; He, X.; Liu, Q.; Ren, S.; Kang, Q.; Zhang, L.; Qu, X., Effect of chromium carbide coating on thermal properties of short graphite fiber/Al composites. Journal of Materials Science 2014, 49, (19), 6705-6715.

[9]    Akbarzadeh, E.; Picas, J. A.; Baile, M. T., Orthogonal experimental design applied for wear characterization of aluminum/Csf metal composite fabricated by the thixomixing method. International Journal of Material Forming 2015, 9, (5), 601-612.

[10]  Liu, T.; He, X.; Liu, Q.; Ren, S.; Zhang, L.; Qu, X., Preparation and Thermal Conductivity of Spark Plasma Sintered Aluminum Matrix Composites Reinforced with Titanium-Coated Graphite Fibers. Advanced Engineering Materials 2015, 17, (4), 502-511.

[11]  Asano, K., Turning Machinability of Short Carbon Fiber Reinforced Aluminum Alloy Composite. Materials Transactions 2016, 57, (8), 1300-1304.

[12]  Asano, K., Mechanical Properties of Aluminum Composites Reinforced with PAN- and Pitch-Based Short Carbon Fibers. Materials Transactions 2017, 58, (6), 906-913.

[13]  Asano, K.; Zainuddin, M. F. B., Wear Behavior of PAN- and Pitch-Based Carbon Fiber Reinforced Aluminum Alloy Composites under Dry Sliding Condition. Materials Transactions 2017, 58, (6), 898-905.

[14]  Baghi, M.; Niroumand, B.; Emadi, R., Fabrication and characterization of squeeze cast A413-CSF composites. Journal of Alloys and Compounds 2017, 710, 29-36.

[15]  Kumar, N.; Chittappa, H. C.; Ezhil Vannan, S., Development of Aluminium-Nickel Coated Short Carbon Fiber Metal Matrix Composites. Materials Today: Proceedings 2018, 5, (5), 11336-11345.

The mechanical properties at high temperature are not described in detail, and creep has not even been cited

Reply: Thanks for your suggestion. In order to describe the tensile properties in detail, we added the tensile tests at 423 K. In addition, we added the stress-strain curves for the tensile tests at room temperature, 423 K and 523 K. We also rewritten the tensile test results. In this work, we put effort on studying the high temperature tensile fracture behavior. The creep test is indeed very important to evaluate the high temperature performance, and we decided to make further research in the next work.

Images should be discussed in greater detail, especially those from TEM. Authors should  use more symbols or text directly within the pictures in order to improve their comprehension

Apply: Thanks very much. Following your suggestion, we added more symbols and text in the TEM pictures to improve their comprehension.

If you have any further questions, please do not hesitate to contact us.

Thank you very much!

Qingnan Meng

College of Construction Engineering

Jilin University, Changchun 130012

R. China

Reviewer 2 Report

1. How were the high temperature tensile tests conducted? how did the authors ensure a constant specimen temperature? was the temperature measured?

The reason for the questions is that some of the reported results are not clear.

For example why is the elongation of pure Al2024 at high temperatures smaller then at room temperature? 

Commonly high temperatures increase aluminum ductility. The authors should either provide a reference of decreased ductility of Al2024 at increased temperatures or offer some other explanation for the results.

2. How did the authors estimate the composite effective possion's ratio? it seems a value of 0.33 (aluminum) was taken in all cases? why is that? how does this assumption influence the estimated elastic modulus?

3. The authors report on an indention test for estimation of the effective elastic modulus. The authors also report that tensile tests were conducted. why is there no comparison between the effective modulus obtained in the tensile testing to that obtained by indention. I would assume that the effective modulus obtained from the tensile tests is more accurate. 

4. It would be worth while if the authors included pictures of the tensile specimens after failure to show the fracture mode obtained in the tensile tests for the different composition's. 

Over all the manuscript well structured and the topic important to study. Nevertheless there are several important issues which must be addressed prior to acceptance. The manuscript should undergo editing by a native English speaker. 

Reviewer 3 Report

48 Some researchers claimed that the Al4C3 formation is adhesive to mechanical properties due 49 to its intrinsic brittleness [29-31].

Re-formulate!

208 Figure 8. (a) Typical load-displacement curves of nanoindentation tests of the SCF and the matrix, (b) 209 the hardness and modulus variation as indention moving from carbon fiber to aluminum matrix

 Re-formulate!

 Probably the images of indents around fibers might be helpful (not obligatory!). The same may be said about "stress-strain" diagram

331 From the result of nanoindentation, the aluminum matrix in the vicinity of carbon fiber is provided with highest modulus (93.53 GPa) and hardness (2.51 GPa). As the indentation moving far away from the carbon fiber, the modulus and hardness for the matrix both  decrease.

Re-formulate!

Author Response

Aug. 14, 2019

Applsci_567818

Title: The evolution of interfacial microstructure and fracture behavior of short carbon fiber reinforced 2024 Al composites at high temperature

Dear Editor and reviewer,

Thank you for the valuable comments. We have addressed all comments from the reviewer as follows:

48Some researchers claimed that the Al4C3 formation is adhesive to mechanical properties due 49 to its intrinsic brittleness [29-31].Re-formulate!

Reply: Following your suggestion, we re-formulate this sentence as ‘Some researchers claimed that the formation of brittle Al4C3 is detrimental to mechanical properties of carbon materials/Al composites [29-31].

208 Figure 8. (a) Typical load-displacement curves of nanoindentation tests of the SCF and the matrix, (b) 209 the hardness and modulus variation as indention moving from carbon fiber to aluminum matrix. Re-formulate! Probably the images of indents around fibers might be helpful (not obligatory!). The same may be said about "stress-strain" diagram

ReplyWe felt sorry for the vague expression of this sentence. We re-formulated this sentence as ‘Figure 8. (a) Typical load-displacement curves of nanoindentation tests of the SCF and aluminum matrix, (b) variation of indentation hardness and modulus of the carbon fiber and adjacent aluminum matrix’. We tried to find the indentation using SEM, but failed because the indentation was too small to be observed. In addition, following your suggestion, we added the stress-strain curves to the text.

331From the result of nanoindentation, the aluminum matrix in the vicinity of carbon fiber is provided with highest modulus (93.53 GPa) and hardness (2.51 GPa). As the indentation moving far away from the carbon fiber, the modulus and hardness for the matrix both decrease. Re-formulate!

ReplyFollowing your suggestion. We re-written the second conclusion as ’From the result of nanoindentation, carbon fiber improves the hardness and modulus of the adjacent aluminum matrix. The indentation on the aluminum matrix nearest carbon fiber possesses the highest modulus (93.53 GPa) and hardness (2.51 GPa).

If you have any further questions, please do not hesitate to contact us.

Thank you very much!

Qingnan Meng

College of Construction Engineering

Jilin University, Changchun 130012

R. China

Round 2

Reviewer 1 Report

Many thanks for following all my suggestion.

when i spoke about creep i was thinking about the mechanism which is active during a high temperature tensile test and i was not referring to the test itself.

anyway now your work is much more solid and ready for publication.

Reviewer 2 Report

The authors answered the questions raised by the reviewer.